

# A low-cost wireless extension for object detection and data logging for educational robotics using the ESP-NOW protocol

Emma I. Capaldi

Phillips Academy Andover, Andover, Massachusetts, United States of America

## ABSTRACT

In recent years, inexpensive and easy to use robotics platforms have been incorporated into middle school, high school, and college educational curricula and competitions all over the world. Students have access to advanced microprocessors and sensor systems that engage, educate, and encourage their creativity. In this study, the capabilities of the widely available VEX Robotics System are extended using the wireless ESP-NOW protocol to allow for real-time data logging and to extend the computational capabilities of the system. Specifically, this study presents an open source system that interfaces a VEX V5 microprocessor, an OpenMV camera, and a computer. Images from OpenMV are sent to a computer where object detection algorithms can be run and instructions sent to the VEX V5 microprocessor while system data and sensor readings are sent from the VEX V5 microprocessor to the computer. System performance was evaluated as a function of distance between transmitter and receiver, data packet round trip timing, and object detection using YoloV8. Three sample applications are detailed including the evaluation of a vision-based object sorting machine, a drivetrain trajectory analysis, and a proportional-integral-derivative (PID) control algorithm tuning experiment. It was concluded that the system is well suited for real time object detection tasks and could play an important role in improving robotics education.

Corresponding author
Emma I. Capaldi,
ecapaldi25@andover.edu

# INTRODUCTION

Machine learning (ML) and artificial intelligence (AI) are changing the way goods and people travel with autonomous vehicles (*Bathla et al., 2022*), the way people write (*Kasneci et al., 2023*), the way people debug programs (*Surameery & Shakor, 2023*), the way people create art (*Mazzone & Elgammal, 2019*), the way teaching is done (*Adiguzel, Kaya & Cansu, 2023*), and the way we work (*Guszcza, Lewis & Evans-Greenwood, 2017*).

Many middle and high schools in the United States have adopted engineering courses that have a robotic component (*Harrell et al., 2004*; *Darmawansah et al., 2023*). In addition, participation in Robotics in middle school and high school has been shown to encourage students to pursue (*Hendricks, Alemdar & Ogletree, 2012*; *Sullivan & Bers, 2019*) and prepare students for *Karim, Lemaignan & Mondada (2015)* careers in STEM. They have also been shown to be effective at improving academic performance (*Jurado-Castro et al., 2023*) and promoting strong computational-thinking (*Evripidou et al., 2020*).

Robotics competitions introduce students to the fields of Mechanical Engineering, Electrical Engineering, Computer Science, and Mathematics in an exciting competitive sports-like environment.

Recently, there has been a push to incorporate AI and machine learning into the middle school and high school curriculum (*Xiong, Wang & Huang, 2018*; *Zhai et al., 2021*; *Dai et al., 2020*; *Knox, 2020*; *Ali et al., 2019*). Project-based learning in robotics has been successful at the university level (*Zhang et al., 2020*) and both game- and project-based learning have been shown to be effective in teaching students about AI (*Leitner et al., 2023*; *Martins & Wangenheim, 2022*). Guiding students through projects involving image recognition is an excellent way to introduce them to the basics of machine learning and prepare them for a world that is quickly integrating AI into everyday life (*Sophokleous et al., 2021*). This article provides a platform and a few examples that can be used to teach project-based AI or ML at the middle school or high school level.

The platform presented integrates with the VEX V5 robotics system. The Robotics Education and Competition (REC) Foundation runs robotics competitions for elementary school, middle school, high school and college students using VEX robotics equipment. The REC Foundation reports that more than 100,000 students participated in 2,600 international events in the competition season of 2022–23. They also claim that 1.1 million students participated in their various programs that distribute educational materials to classrooms (*REC Foundation, 2023*). The VEX robotics system used in competitions has been purchased and used by many high schools throughout the United States and abroad and can be used beyond the competitions to teach STEM concepts through hands-on learning activities. However, the VEX robotics hardware includes a rudimentary vision sensor capable of detecting competition objects by color at 50 fps, and the Cortex A9 microprocessor running the competition equipment has limited processing capability and a fixed ecosystem of plug-and-play peripherals. It cannot run complex machine learning models without additional computational power.

In order to run AI or ML applications, the VEX equipment must be paired with a more capable processor and camera. There has been some success pairing the previous generation VEX Cortex microprocessor with a Raspberry Pi (*He & Hsieh, 2018*), and pairing the VEX V5 microprocessor with a Raspberry Pi (*Zietek et al., 2022*). The Raspberry Pi is a capable processor that has been used for many IOT applications (*Nguyen et al., 2022*) and even used to run convolutional neural networks (*Sabri & Li, 2021*). However, the Raspberry Pi requires an additional power source and lacks the computational power to execute complex vision models. In this article, the VEX V5 microprocessor is paired with an ESP32 to communicate wirelessly with a computer system. ESP32s are inexpensive, widely available microcontrollers that communicate wirelessly with each other over long distances using the ESP-NOW protocol. The low overhead and fast through rate for the protocol make it fast enough to be used in real-time robotic control applications. ESP32s have already been used for a wide range of projects, such as the remote monitoring of bee colonies (*Kviesis et al., 2023*), environmental

monitoring (*Winkler, 2021*), air quality monitoring (*Truong, Nayyar & Masud, 2021*) and low-energy wireless networks (*Labib et al., 2021*). Others have also incorporated OpenMV (*Abdelkader, El-Sonbaty & El-Habrouk, 2017*) machine vision modules, which integrate a camera with a basic low-power microprocessor for vision applications (*Guo, Wu & Fang, 2020*; *Wei-Peng et al., 2020*).

Educational robotics also benefits from real-time data visualization. Logging and visualization can improve students' investigative, analytical, and interpretive skills (*McFarlane & Sakellariou, 2002*), improve student understanding of key physics concepts (*Alimisis & Boulougaris, 2014*), and change the way students think about experiments (*Barton, 2004*). The access in the Python programming language to libraries for data analysis, visualization, computer vision, and machine learning makes it well suited for this task (*Fraanje et al., 2016*).

This article outlines the construction, coding, and usage of a low-cost communication interface between the widely available VEX V5 microprocessor, a computer system, and a camera system to implement machine learning algorithms and data log in educational environments. The bidirectional system allows data transfer from the robotic system to the visualization system and commands to be sent from the computer system to the robotic system. Due to the increased processing capability, this system extends the functionality and usage cases for the VEX V5 system, allowing students to run more advanced code on their robot. Wireless communication between ESP32s also gives students real-time access to data streams from their robot. This article discusses possible ways in which this data could be used to introduce students to important and interesting topics in robotics, such as proportional-integral-derivative (PID) controller optimization and trajectory analysis. The study also integrates a widely available OpenMV camera, which provides both a photodetector and an onboard microprocessor. This gives the system the ability to run real-time image recognition programs and provides an excellent platform for teaching students about object detection algorithms.

## MATERIALS AND METHODS
### Hardware configuration
#### ESP-NOW
The ESP32 (*Espressif Systems, 2022*) supports a proprietary communication protocol called ESP-NOW that utilizes the 2.4 GHz band and is optimized to send short messages of up to 250 bytes to other ESP32 nodes with very little overhead, a quick response time on the order of one millisecond, low power usage, and a range between nodes of over 200 m (*Yukhimets, Sych & Sakhnenko, 2020*). These characteristics make the ESP32 ideal for reading data from and sending instructions to real time control systems and data logging (*Linggarjati, 2022*). The onboard ESP32 does not require a separate battery and can be powered by the output voltage on any of the VEX robot three-wire ports which support an output of 5 volts at a maximum of 2 amps. The ESP-NOW link was operated at a bit rate of 1 Mbps.

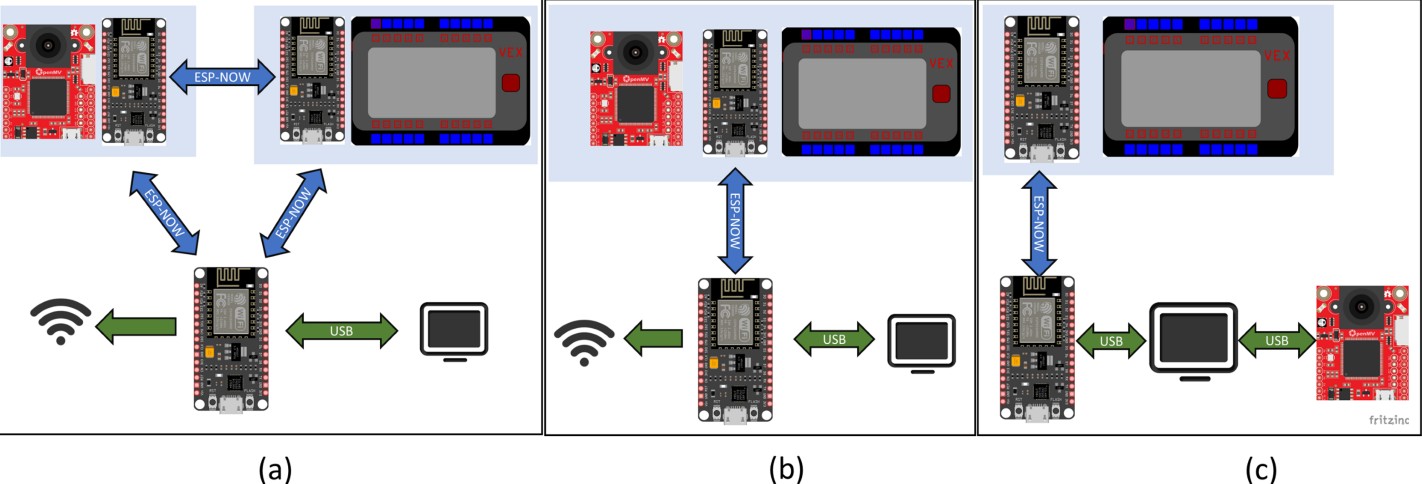

**Figure 1 Potential configurations for the remote ESP-NOW enabled connection between the computer, OpenMV camera, and the VEX V5 microprocessor.** (A) OpenMV camera is directly connected to the computer *via* a USB cable while the computer and VEX V5 microprocessor communicates *via* ESP-NOW. (B) The OpenMV camera and the VEX V5 microprocessor communicate with the computer through a single ESP-NOW link. (C) The OpenMV camera and the VEX V5 microprocessor are each linked to a separate ESP32 that allows communication through ESP-NOW between the camera, the VEX V5 microprocessor and the computer. In all cases, one ESP32 was connected to the VEX processor and one ESP32 was hardwired to the computer.

The ESP32 also has Wi-Fi communication built into the board. Wi-Fi communication would work well for the transmission of data from the board to a computer for applications such as data logging. However, the overhead associated with this communication would make it ill-suited for usage in a real-time control system where the external computer is driving the robot and responding to sensor data transmitted by the robot.

### OpenMV

The H7 OpenMV camera (*Abdelkader, El-Sonbaty & El-Habrouk, 2017*) module used in this study has an ARM 32-bit Cortex M7 CPU and consumes 140 mA when active. It can be powered using the five-volt three-wire port power on the VEX V5 microprocessor. The USB connection between the camera and the computer can transmit data at rates up to 12 Mb/s. The camera can take 16-bit RGB565 images at a resolution of 640 × 480 at 75 frames per second. The onboard processor is capable enough to run small tensorflow lite models, run the opencv image processing library, and perform basic image pre-processing if needed. Programs can be written for and run on OpenMV hardware using Python.

### Connectivity

Three different hardware configurations were envisioned for this system, as shown in Fig. 1. Each of these configurations has strengths that would make it useful for different applications.

The first configuration, Fig. 1A, has the VEX V5 microprocessor connected to an ESP32 with the OpenMV camera, computer, and another ESP32 connected to each other. Direct

connection of OpenMV to the computer *via* USB supports the rapid transmission of images. This is useful for real time computationally intensive image analysis. Commands are then sent to the robot *via* the ESP-NOW connection. This configuration is suitable for a case where the robot actuation is occurring in response to a visual que in a fixed region, such as machine automation tasks and quality control. This first configuration was selected for the sorting experiments detailed later in this article.

The second configuration, Fig. 1B, has the OpenMV and VEX V5 microprocessor connected to the same ESP32. The VEX V5 is connected *via* a RS485 bridge while the OpenMV camera is connected *via* a serial connection. The vision analysis would be limited to tensor flow lite models, which can be run directly on the OpenMV camera. This configuration has the advantage that the OpenMV camera, the VEX V5 microprocessor, and the ESP32 can be powered using the VEX battery.

The third configuration, Fig. 1C, has the VEX V5 microprocessor, the computer, and the OpenMV camera, each connected to one of the two independent ESP32s. If the camera and the robot were both remotely located and the camera was not traveling with the robot, this configuration might be considered. This would be suitable for a case where a tensor flow lite model can be used and run directly on the OpenMV camera, negating the need for heavy computation on the computer.

All three of these configurations are viable and benefit from the added computational power gained by having a computer in the loop. In addition, each ESP32 can be used to interface with additional sensors such as lidar, temperature, or GPS sensors *via* the I2C protocol. These types of sensors are unavailable through the official VEX manufacturer, and I2C is unavailable on the VEX V5 microprocessor. These additional sensor types would greatly increase the scope and type of experiments that could be conducted using the VEX V5 microprocessor.

### Wiring

The VEX V5 microprocessor has 21 smart ports, of which 20 are located on the front of the device as shown in Fig. 2. These smart ports can communicate with external devices using the RS485 standard and provide power at 12 Volts. The VEX V5 microprocessor also has eight three-wire ports located on the side of the device, which can be used to supply power at 5 Volts. The ESP32 power input pin is connected to a 5 Volt power supply *via* the three-wire port on the VEX V5 microprocessor. The RS485 signal emitted through the smart port on the VEX V5 microprocessor is decoded by a MAX485 chipset. Power to the RS485 decoder is supplied *via* the 3.3 V power output pin on the ESP32. The RS485 decoder used in this study can be set to transmit or receive a signal based on the state of the DE and RE pins. Bidirectional communication is achieved by driving these pins using the ESP32. When a signal is received *via* the ESP-NOW protocol, the ESP32 sets the state of the RS485 to transmit mode. A message is then sent *via* RS485 to the VEX V5 microprocessor. The ESP then sets the RSS485 decoder back to receive mode so that the VEX V5 microprocessor may transmit messages through the ESP32 back to the computer system. The wiring diagram showing the connection between the ESP32, RS485 decoder, and VEX V5 microprocessor is shown in Fig. 2.

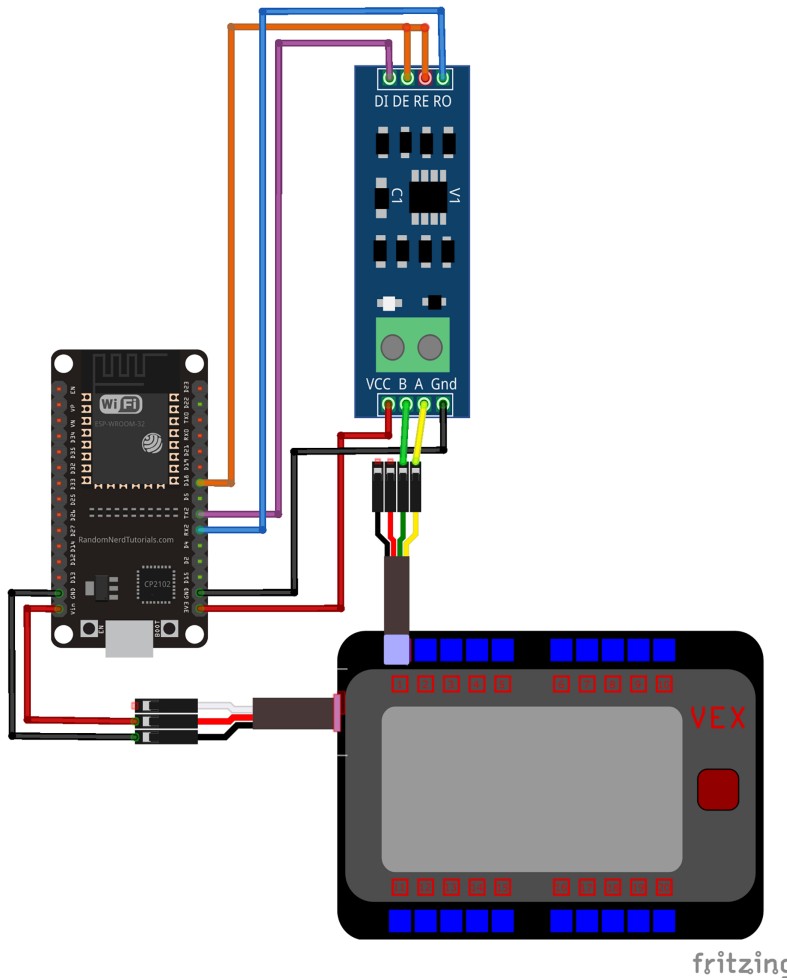

**Figure 2 Wiring diagram between ESP32, RS485 decoder and VEX V5 microprocessor.** As shown, the ESP32 is connected to a three wire port on the microcontroller which supplies power. The RS485 decoder receives and transmits signals to the microcontroller *via* the smart port.

## Software

Three distinct pieces of software were written and are available in the supplemental data section of this article. This includes a driver for communication between the two ESP32s, a code for running the VEX V5 microprocessor, and a GUI for use on the computer system.

### ESP32 communication code

The Arduino IDE was used to code a communication link between two ESP32 microcontrollers using the ESP-NOW communication protocol. The ESP32 connected to the VEX V5 microprocessor continuously transmits system status data *via* the ESP-NOW protocol at a fixed interval to another ESP32. The ESP32 connected *via* USB to the computer receives the data and repeats them to the computer *via* a serial link. Commands sent by the computer through the USB are echoed in the opposite direction back to the ESP32 connected to the VEX V5 microprocessor.

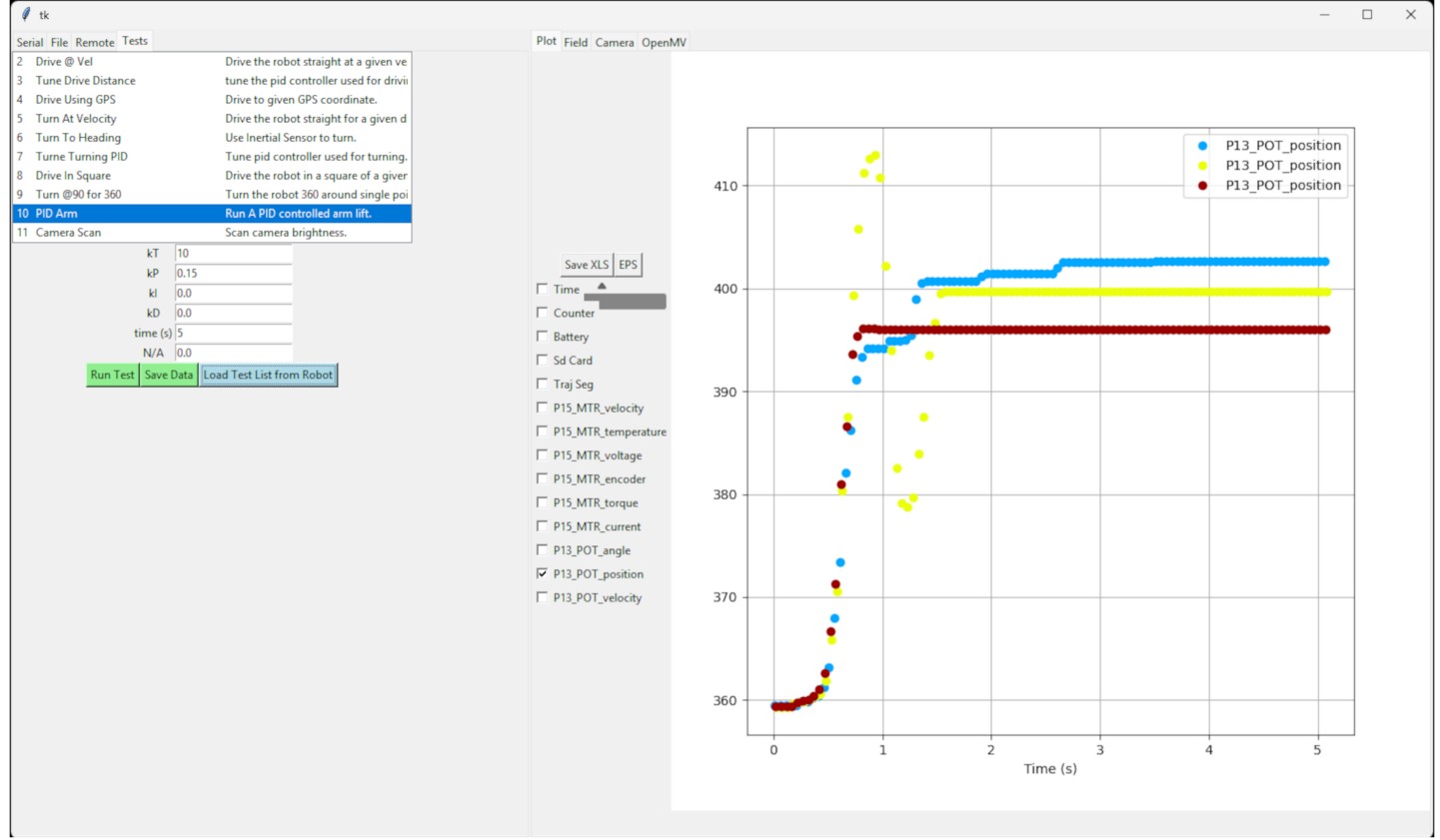

**Figure 3** Screenshot showing the Python GUI running on the computer system which can be used to send commands wirelessly *via* the ESP-NOW connection to the robot and to display data in real time from sensors and motors on the VEX V5 microprocessor. Here the results of multiple runs are displayed simultaneously and can be compared with one another.

### VEX code

A code template written in C++ is provided which runs the communication link on the VEX V5 Microprocessor. It provides a starting point for users interested in modifying and extending the capabilities of this system. The code automatically detects the presence of VEX peripherals and transmits data from each of these peripherals at a regular interval to the computer. The code also provides a mechanism for the VEX V5 microprocessor to recognize custom commands sent from a computer and to run predefined tests in response to computer instruction. The code can be edited using Visual Studio Code with the VEX *VS* Code Extension installed and then downloaded to the VEX V5 microprocessor.

### Python GUI

A graphical user interface, Fig. 3, was written in python using the Tkinter graphical user interface (GUI) library. This library is included in the standard installation of Python. The data received from the VEX V5 microprocessor are plotted in real time using MatplotLib (*Hunter, 2007*). This data can be stored or retrieved from files on the hard drive, plotted, and analyzed using the python code. The python code can also capture and display images

**Table 1 Device cost breakdown as determined on Amazon.com on 6/14/2023.**

| Item | Quantity | Cost (in US$) |
| --- | --- | --- |
| ESP-32S wifi development board | 2 | $5.67 |
| TTL to RS-485 MAX485 module | 1 | $1.48 |
| 2.4 × 1.4 × 1 inches electronics case | 1 | $1.44 |
| 2.36 × 1.42 × 0.67 inches electronics case | 1 | $1.40 |
| OpenMV camera module | 1 | $82.00 |
| Total | | $97.66 |

from an OpenMV camera connected *via* USB to the computer. It can be used to perform machine learning operations on the images and send commands back to the VEX V5 microprocessor. Users can modify the code to create hybrid control systems that are running some processing on the computer platform and some on the VEX V5 microprocessor while sharing data and commands between the two. The Python multiprocessor module is used to isolate GUI operations from data transmission, resulting in a smooth GUI and much improved responsiveness when compared to a single-processor algorithm.

## Cost

The system hardware was designed with minimal cost and footprint. It is intended for use in high schools that already have a VEX program in place. A VEX starter kit costs on the order of $1,000 USD. The parts needed to extend the system to use machine learning and sophisticated machine vision cost less than $100 USD, as shown in Table 1. The ESP32/RS485 package connected to the VEX V5 microprocessor fits within an electronics case with dimensions 2.4 × 1.4 × 1 inches. The ESP32 connected to the computer fits within an electronics case with dimensions of 2.36 × 1.42 × 0.67 inches.

## RESULTS AND DISCUSSION

### Message transit time

The time required for a message to be sent from the computer to the VEX V5 microprocessor, processed by the VEX V5 microprocessor, and then for a response to be sent back to the computer was determined. Since the message round-trip starts and ends on the same computer, the round-trip time can be computed by subtracting the initiation time from the termination time stamp. The mean round trip message time was determined from 10 trials to be 187 ms with a standard deviation of 57 ms.

Additionally, the roundtrip path taken by the data message from the computer to the VEX V5 was analyzed. Both the ESP board and the VEX V5 microcontroller have digital output pins, which can be monitored externally *via* an oscilloscope. The programs running on each ESP32 microcontroller were modified to set the LED on pin 2 of each ESP32 to high when messages are first detected on the input line and set the pin back to low when

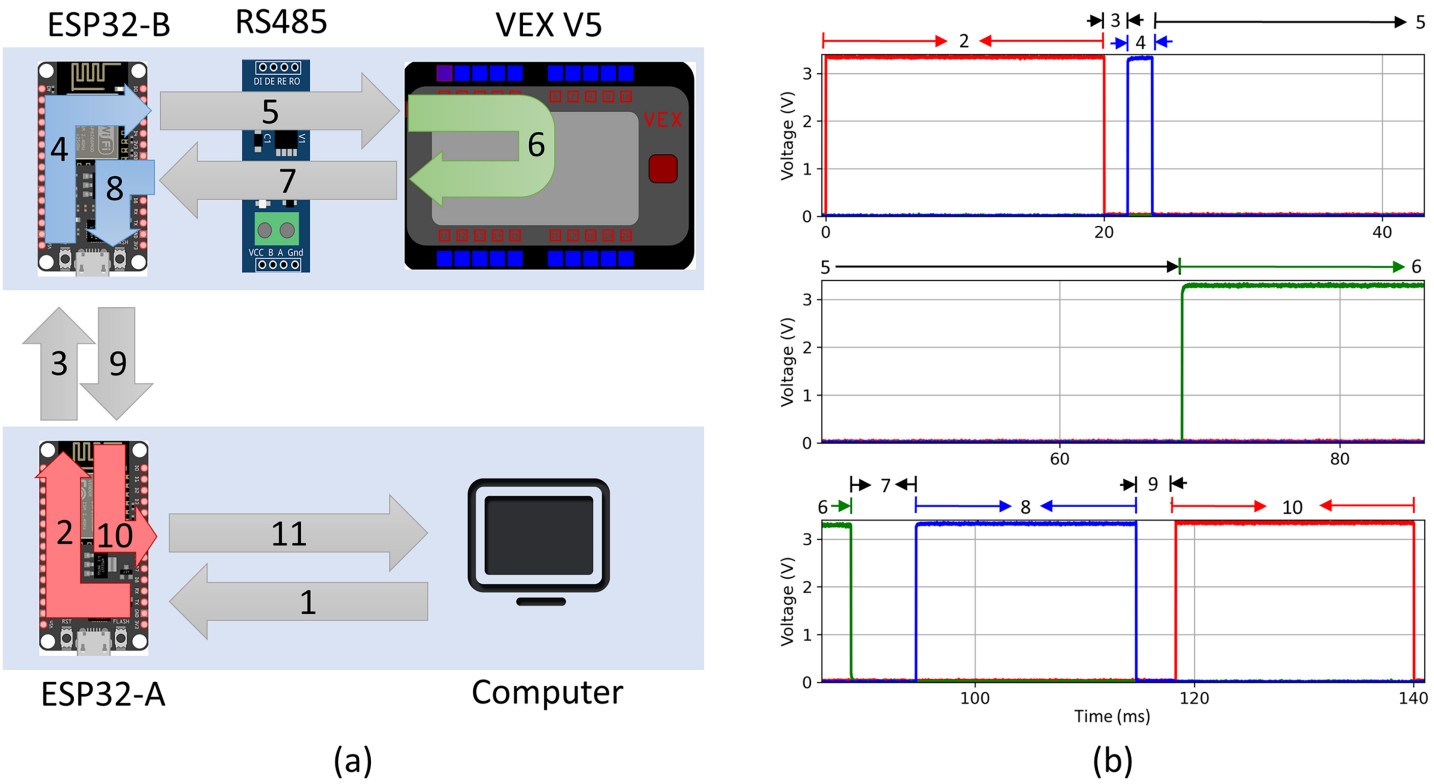

**Figure 4 (A) Illustration of message path from computer to VEX V5 and back. (B) Characteristic signal recorded using four channel oscilloscope.**

the message has been sent on the output line. The program running on the VEX V5 microprocessor was similarly modified to set the digital output on port B to high when a message was detected and back to low after it was echoed onto the serial line. This allowed us to track the detailed progress of the message through the path shown in Fig. 4A. A four channel InfiniVision MSOX4154A mixed signal oscilloscope was used to simultaneously monitor all three microcontrollers. The output signal over time was used to determine precisely when each message was detected and transmitted. Channel 1 of the oscilloscope was connected to pin 2 on ESP32A, channel 3 was connected to pin 2 on ESP32B, and channel 4 was connected to the digital output port B on the VEX V5 microprocessor. This allowed us to calculate detailed time intervals for each of the path segments in Fig. 4A. Figure 4B shows a characteristic oscilloscope reading for one of the 12 trials conducted, labeled with each corresponding segment of the pathway. The mean and standard deviation for each segment were calculated from the twelve oscilloscope readings, as shown in Table 2. Segments 1 and 11 representing the transit time between the computer and ESP32-A could not be determined directly. Instead, they were computed by subtracting the direct segment time measurements for 2 through 10 from the round trip message time and dividing by two under the assumption that the time of segment 1 is equal to segment 11.

**Table 2 Average and standard deviation of time taken for message to traverse given path segment (*n* = 12).** The message path is graphically illustrated in Fig. 4A.

| Segment | Description | Mean [ms] | STD [ms] |
|---|---|---|---|
| 1* | Message sent from python GUI to ESP32-A *via* USB serial connection. | 29.8 | – |
| 2 | ESP32-A recieves message *via* USB serial connection (pin set to high) and then echoes message to ESP-B *via* ESP-NOW (pin set to low). | 20.0 | 0.0012 |
| 3 | Message is sent from ESP32-A to ESP32-B *via* ESP-NOW. | 2.01 | 0.819 |
| 4 | ESP32-B recieves message from ESP32-A *via* ESP-NOW (pin set to high) and then echoes message to RS485 *via* serial connection (pin set to low). | 1.56 | 0.361 |
| 5 | RS485 recieves message from ESP32-B *via* serial connection and transmits converted message to VEX V5 microprocessor. | 34.5 | 11.4 |
| 6 | VEX V5 microprocessor recieves message, processes response, and sends reply to RS485. | 19.9 | 0.0105 |
| 7 | RS485 recieves message from VEX V5 microprocessor and transmits converted message to ESP32-A *via* serial connection. | 4.46 | 2.24 |
| 8 | ESP32-B recieves message from RS485 *via* serial connection (pin set to high) and echoes message to ESP32-A *via* ESP-NOW (pin set to low). | 20.1 | 0.004 |
| 9 | Message is sent from ESP32-B to ESP32-A *via* ESP-NOW. | 3.19 | 0.663 |
| 10 | ESP32-A recieves message *via* ESP-NOW (pin set to high) and then echoes message to python GUI *via* USB serial connection (pin set to low). | 21.7 | 0.0037 |
| 11* | Message sent from ESP32-A to python GUI *via* USB serial connection. | 29.8 | – |

**Note:**
* The time for segments 1 and 11 was computed by subtracting the average total time for segments 2 through 10 from the roundtrip time and dividing by 2.

**Table 3 Average and standard deviation of the percent of data messages lost between transmitter and receiver outdoors as a function of distance.**

| Distance (m) | Mean [%] | Standard deviation [%] |
|---|---|---|
| 0 | 0.00 | 0.00 |
| 10 | 0.197 | 0.139 |
| 20 | 0.571 | 0.410 |
| 30 | 0.861 | 1.22 |
| 40 | 0.126 | 0.178 |
| 60 | 13.2 | 10.4 |

## Packet loss

The practical distance limits between the computer and the VEX V5 microprocessor described in this experiment were evaluated by determining the wireless packet loss as a function of distance. A VEX V5 microprocessor which was configured to transmit data at a regular interval of 50 ms was attached to a drivetrain. The system was then placed outdoors on the sidewalk and a laptop with a receiving ESP32 connected to it was moved at intervals of 10 m away from the robot. A clear line of sight was maintained between the transmitter and receiver. Data was transmitted from the VEX V5 microprocessor at intervals of 50 ms

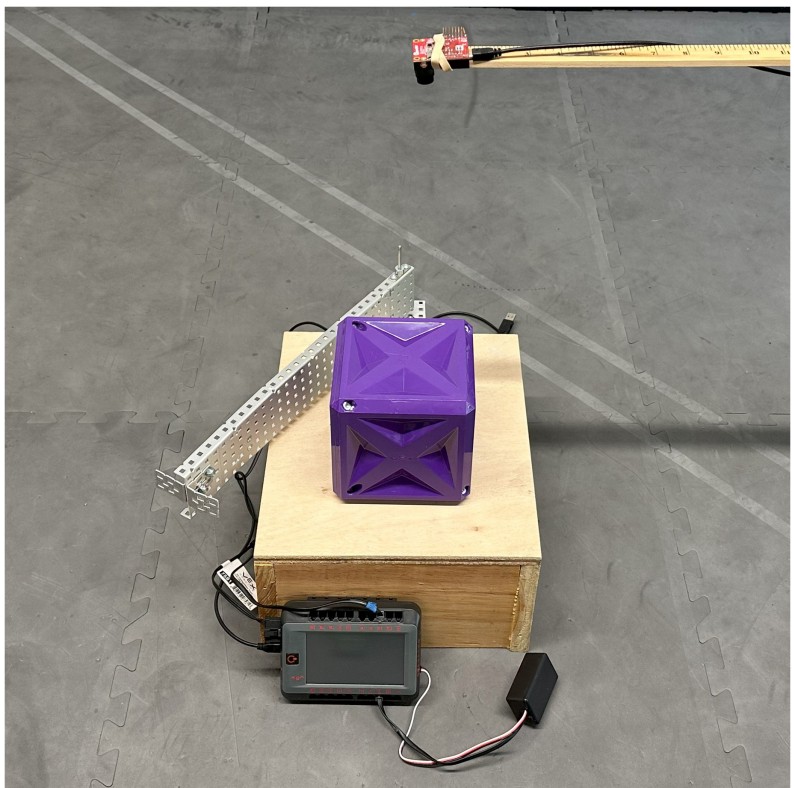

**Figure 5 Simple sorting machine used to test visual recognition of VEX game pieces.** Objects are pushed to the right or the left depending on their identified type.

and received on the computer end. The number of data packets received by the computer in a 15 s interval was recorded. Packet loss, *PL*, is calculated as follows:

$$PL = \frac{n_e - n_a}{n_e} \times 100\% \qquad (1)$$

where $n_e$ is the number of expected packets received from the ESP-32, and $n_a$ is the number of packets actually received. The mean and standard deviation of the percent loss for each distance interval are shown in Table 3. The packet loss was less than 1% between 0 and 40 m. For data logging, the functionality is sufficient up to 60 m, but for control applications where the computer is in the control loop, the distance between transmitter and receiver should be 40 m or less.

## Machine learning sorting application

As an example of an application that uses machine vision to identify objects and drive the motion of a physical system, a simple sorting experiment was created. As shown in Fig. 5, the sorting machine has an OpenMV camera placed above a platform on which VEX game pieces are placed. The machine can then sort the pieces to the left or right side of the platform.

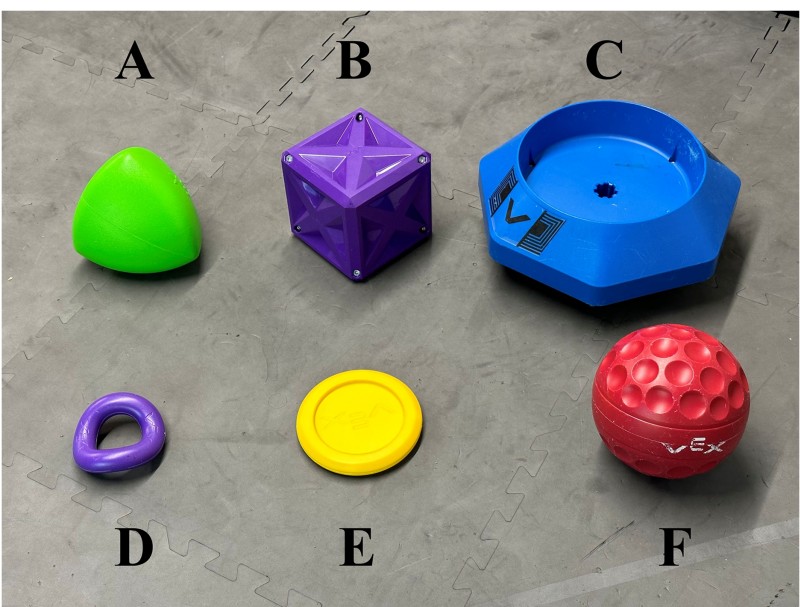

**Figure 6 VEX game pieces used for sorting experiment.** The object types include the (A) acorn, (B) box, (C) container, (D) ring, (E) disc, and (F) ball.   

VEX game pieces were collected from the last five game seasons, as shown in Fig. 6. These game pieces differ significantly from each other and make a good proof-of-concept experiment. An OpenMV H7 camera was held 19 inches above a platform onto which the VEX competition objects were placed. The camera was used to obtain 640 × 480 pixel RGB images of objects on the platform. The images from the camera were analyzed by a computer, the YoloV8 algorithm was used to determine the object type, and then a signal was sent to the VEX V5 microprocessor. In response to signals from the microprocessor, a motor swings an arm across the platform, sweeping objects either to the left or to the right. The system can be programmed to sort one VEX game object type to the left and another VEX game object type to the right. To test the accuracy of the model, we programmed the system to swing the arm to the left if a box was detected, to the right if an acorn was detected, and not to move the arm if no object was detected. Objects were placed on the platform or the platform was left empty for 2 s intervals to allow the arm an appropriate time to react. Each object was placed on the platform 10 times, for a total of 30 trials. A random number generator was used to determine the order in which objects would be placed on or removed from the platform.

## Training

The YOLO (You Only Look Once) v8 image object identification algorithm was selected for use in this project because of its ability to identify objects with high accuracy at high speed. The model was trained on images of VEX game objects from multiple years of competitions, as shown in Fig. 6. A set of 116 images was taken at a size of 640 × 480 pixels using the OpenMV camera of the game objects in various positions, orientations,

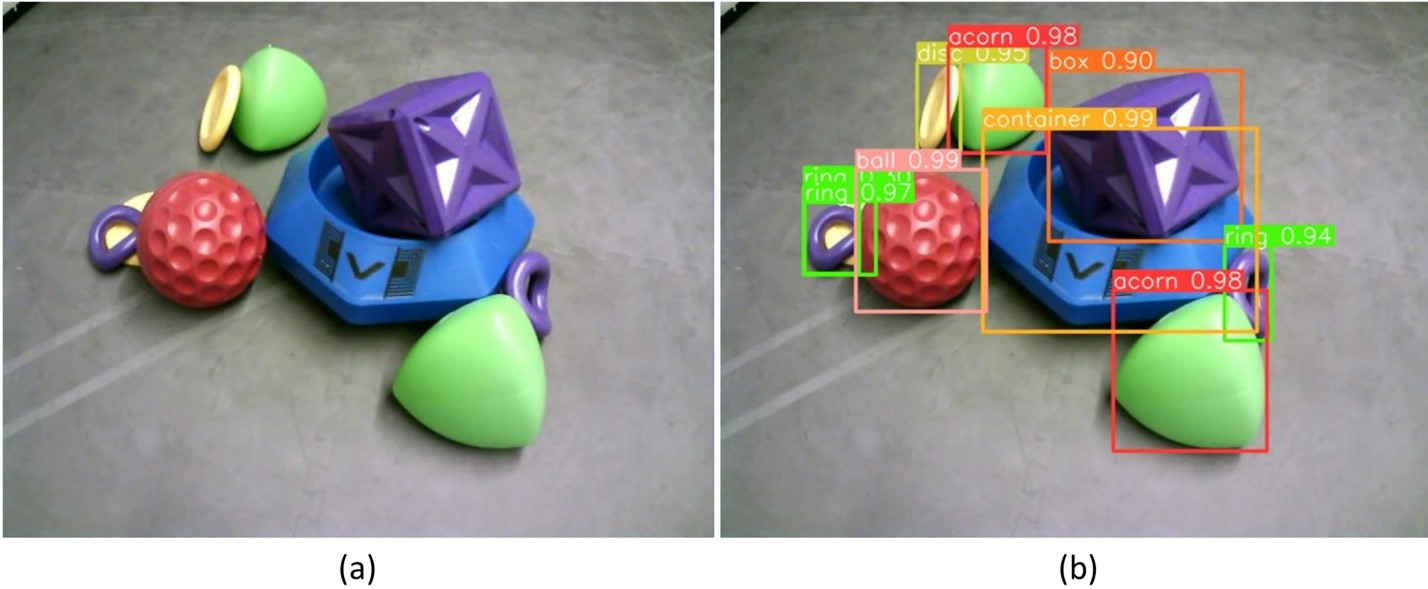

**Figure 7** (A) Raw image taken using OpenMV H7 camera at 640 × 480 resolution of partially occluded VEX game pieces. (B) Predicted object labels using YOLOV8 algorithm and trained weights.

groupings, and at various angles. Figure 7A shows one of the images that were taken. Training images were imported within LabelStudio (*Tkachenko et al., 2020–2022*), an image labeling software that makes it easy to label objects for image recognition algorithms. Each distinct VEX game object was assigned a tag. For each image, all identifiable objects were assigned a tag and labeled with a bounding box. After the images were labeled, they were divided into training, testing and validation sets, which consisted of 86, 16, and 14 images, respectively. Training was run for 50 epochs and the best set of YoloV8 parameters from these 50 epochs was used as the final model.

After training was completed, the accuracy of the best resulting YoloV8 model was measured using the validation set of images. Figure 7B illustrates how the model can label identified objects within an image. The name of each object type and the confidence for each prediction is shown at the top of the bounding box for each object. The sample image illustrates that most objects can be identified with high confidence (above 90%). It also shows that some objects can be missed, such as the yellow disc shown behind a purple ring, and some objects can be identified twice, such as the double bounding boxes around the purple ring. To quantify these errors and labeling performance, the precision, recall, mAP, and processing time of the model were evaluated.

## Precision and recall

Precision and recall are useful values to evaluate the performance of a machine learning model. Precision, $P$, is a measure of how many identifications made were correct, while recall, $R$ is a measure of how many objects were identified correctly, and are defined as follows:

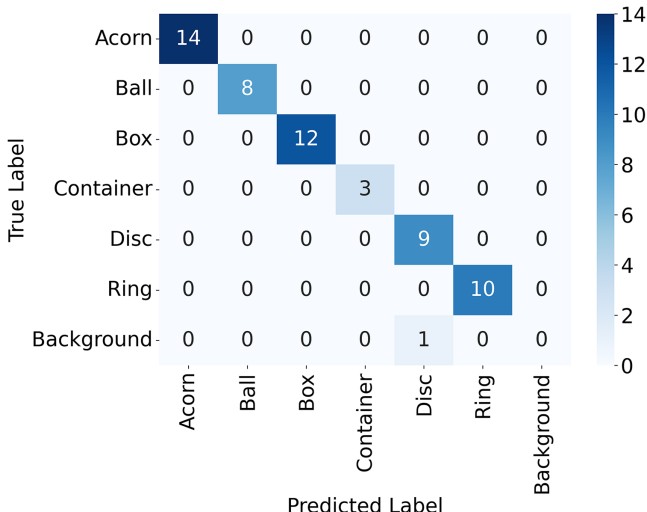

**Figure 8  Confusion matrix for the objects detected in the validation image using YoloV8.**

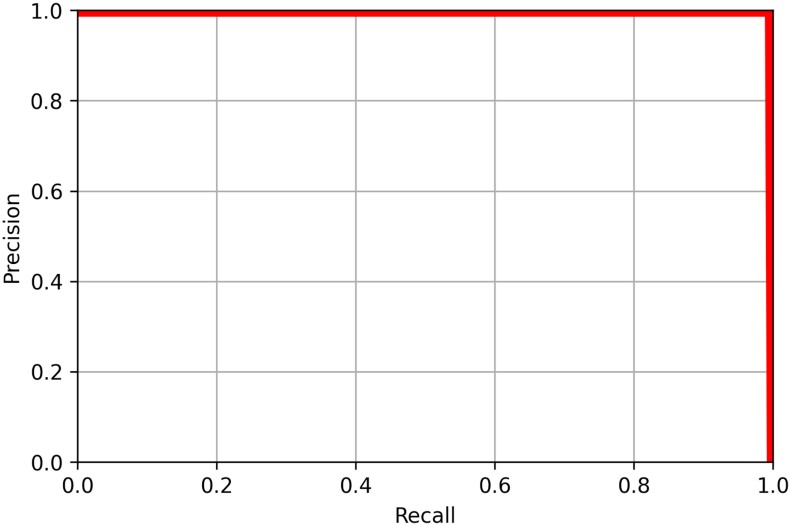

**Figure 9  The precision-recall curve for all classes of objects detected in the validation image using YoloV8.** The value of mAP was 0.995 at a recall of 0.5.

$$P = \frac{TP}{TP + FP} \tag{2}$$

$$R = \frac{TP}{TP + FN} \tag{3}$$

where $TP$ refers to the number of true positives, the number of labeled objects that were correctly identified, $FP$ to the number of false positives, the number of predicted objects that did not exist, and $FN$ to the number of false negatives, the number of labeled objects that were not identified. $TP$, $FP$, and $FN$ are calculated from the confusion matrix, a table that shows the number of each object that was labeled as each of the possible object labels,

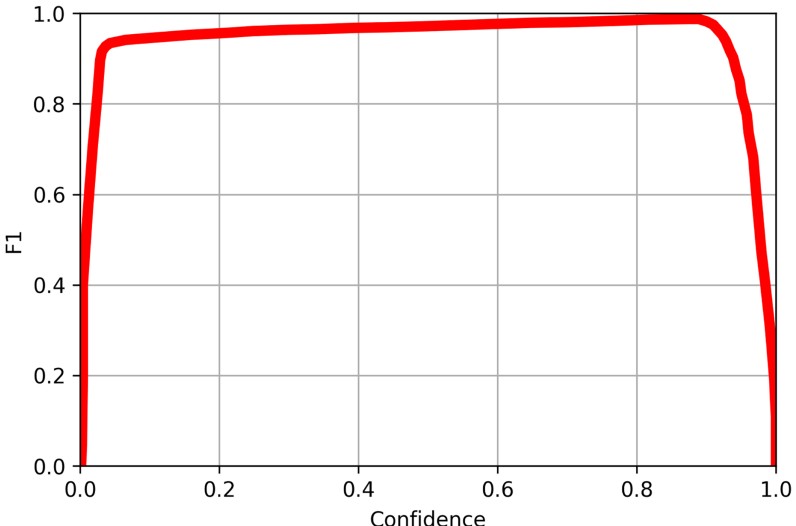

**Figure 10** **The F1 curve for all classes of objects detected in the validation image using YoloV8.** The F1 value reached 0.99 at confidence level of 0.882 for all classes.

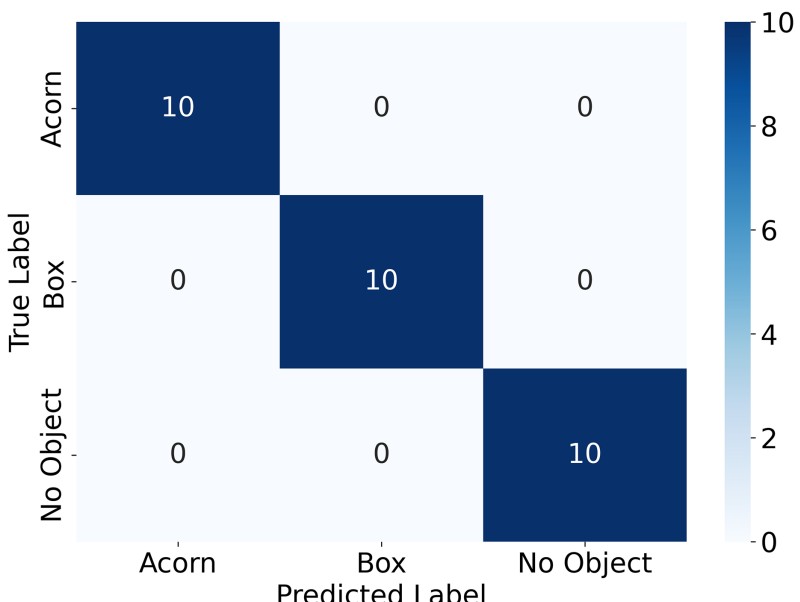

**Figure 11** **The confusion matrix obtained from the object sorting experiment.**

as shown in Fig. 8. The precision and recall scores are between 0 and 1, and a more accurate model has both a higher precision score and a higher recall score. The precision-recall curve for each of the categories of objects tested is shown in Fig. 9. All objects had precision and recall scores close to 1, demonstrating that the model correctly identified objects most of the time. Precision and recall values can also be combined to generate an F1 score, another common way to evaluate the performance of a model. The F1 score is defined as follows:

**Table 4 Metric results obtained from the object sorting experiment.**

| Metric | Value [%] |
|---|---|
| Recall (ball) | 100 |
| Precision (ball) | 100 |
| Recall (acorn) | 100 |
| Precision (acorn) | 100 |
| Accuracy | 100 |

$$F1 = \frac{2 \times P \times R}{P + R} \qquad (4)$$

F1 scores lie between 0 and 1, and a more accurate model has a higher F1 score. The F1 scores for all object category was used to generate an F1 curve, as shown in Fig. 10. All classes had F1 scores close to 1 at most confidence levels, and all classes had an F1 above 0.99 at a confidence level of 0.882.

A confusion matrix, Fig. 11, and precision, recall, and accuracy scores were also calculated for the sorting experiment described earlier. The results are shown in Table 4. The system correctly identified both objects and an empty platform with 100% accuracy.

## mAP score

Another value of interest for evaluating the performance of the model is the mean average precision value, *mAP*, which is equal to the area enclosed by the precision-recall curve and the coordinate axis. It is calculated as follows:

$$mAP = \frac{\sum_{k=1}^{k=n} AP_k}{n} \qquad (5)$$

where $n$ is the number of classes in which the model is trained, and $AP_k$ is the average precision of the class $k$. mAP scores lie between 0 and 1, and a more accurate model has a mAP score closer to 1. The difference between a mAP50 and mAP50-95 score lies in the way that true and false positives are identified. A label is classified as true positive when a certain percentage of the labeled area overlaps with the true object boundary. If the overlapping area does not meet this threshold, it is labelled as a false positive. This overlap area is known as the intersection over union, or IoU value. To calculate the mAP50 score, the IoU value of a predicted object must be greater than or equal to 0.5 to be classified as a true positive. To compute the mAP50-95 score, the number of true positives is computed as the average of the number of true positives for each IoU threshold value between 0.5 and 0.95. The mAP50 and mAP50-95 values are shown in Table 5, and are all close to 1. The mAP50 scores are larger than the mAP50-95 scores, which is expected, as there should be fewer true positives as the IoU threshold value increases.

**Table 5 mAP values computed from validation training set of 14 randomly selected images using the weights from the best set of weights obtained after 50 training epochs.** The column labeled instances gives the number of labeled objects of each type found within the set of 14 images.

| Class | Instances | mAP50 | mAP50-95 |
| --- | --- | --- | --- |
| All | 57 | 0.955 | 0.924 |
| Acorn | 14 | 0.995 | 0.940 |
| Ball | 8 | 0.995 | 0.941 |
| Box | 12 | 0.995 | 0.940 |
| Container | 3 | 0.995 | 0.995 |
| Disc | 10 | 0.995 | 0.865 |
| Ring | 10 | 0.995 | 0.863 |

**Table 6 Average inference times obtained by executing the YoloV8 algorithm on various hardware platforms.**

| Device | Type | Preprocessing mean (stdev) [ms] | Inference mean (stdev) [ms] | Postprocessing mean (stdev) [ms] |
| --- | --- | --- | --- | --- |
| AMD Ryzen 9 5950X | CPU | 1.09 (0.325) | 62.8 (2.49) | 0.53 (0.450) |
| NVIDIA Quadro P2200 | GPU | 0.750 (0.493) | 8.41 (1.67) | 1.64 (0.585) |
| Apple M2 | CPU | 0.772 (0.351) | 52.1 (14.4) | 0.472 (0.957) |

## Inference time

The YoloV8 algorithm takes an image as input and then conducts a preprocessing step that resizes the image, pads the image to have a square shape, normalizes the pixel values, and converts the pixel array to a pytorch tensor. The inference step identifies objects within the processed image, and then post-processing conducts a non-maximal suppression (NMS) step. The processing time for each of these steps was recorded. Table 6 shows the time taken for preprocessing, inference, and postprocessing for a single image on a set of devices with the algorithm running on the CPU or GPU averaged over the set of validation images.

## Real time data logging

### Trajectory analysis

Repeatability and reliability are a recurrent problem in middle- and high-school robotics competitions and in the educational curriculum. Understanding how design choices and control algorithms influence the reproducibility of a particular motion is particularly useful. In this section, a simple six wheel drivetrain powered by two motors, as shown in Fig. 12A, is created and outfitted with a VEX GPS sensor. The VEX GPS sensor allows the drivetrain's position to be determined within a competition field. Though the name implies this sensor relies on GPS, it is in fact vision based and uses images of an encoded strip placed along the field perimeter to determine the location.

The drivetrain was programmed to travel around the field in a prescribed square pattern, as shown in Fig. 12B with segment definitions described in Table 7. Tests are triggered by the GUI that runs on the computer and are parameterized, allowing students

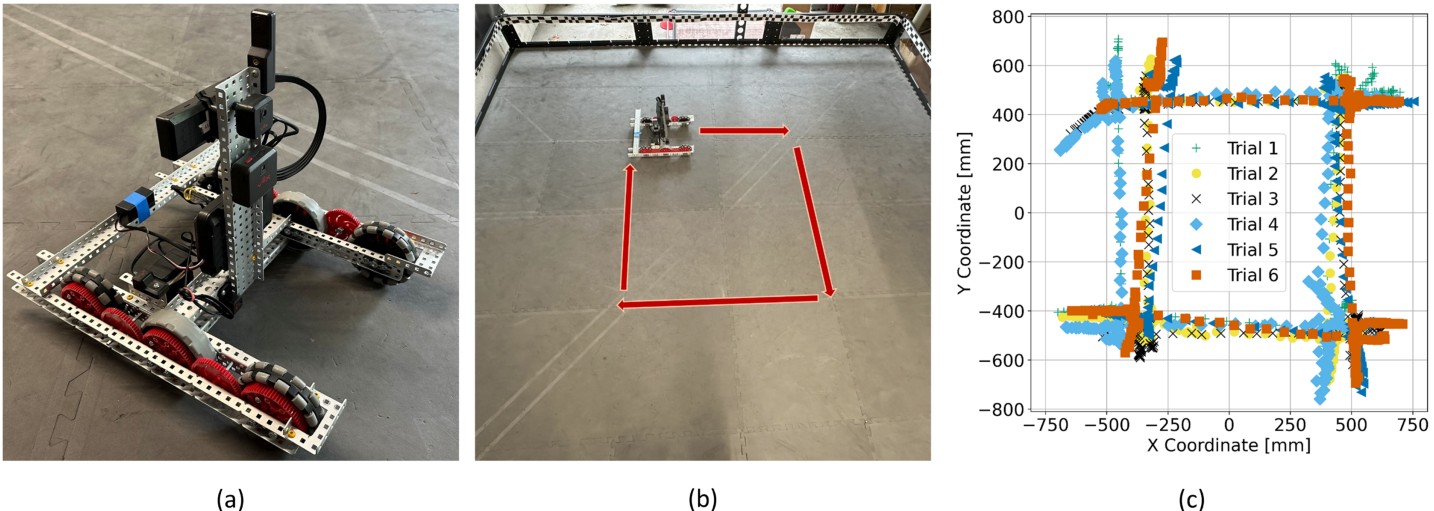

**Figure 12** (A) Drivetrain used in testing. The ESP32 transmitter is located on the back of the drivetrain held in place with blue tape. (B) Photograph of the VEX field taken from above with the robot path labelled with red arrows. (C) Several trial trajectories collected as the robot travels the prescribed path.

**Table 7 Drivetrain movement segment definitions.**

| Segment ID | Description |
|---|---|
| 0 | Starting position |
| 1 | Drive forward 40 inches (1,016 mm) |
| 2 | Turn right by 90 degrees |
| 3 | Drive forward 40 inches (1,016 mm) |
| 4 | Turn right by 90 degrees |
| 5 | Drive forward 40 inches (1,016 mm) |
| 6 | Turn right by 90 degrees |
| 7 | Drive forward 40 inches (1,016 mm) |
| 8 | Turn right by 90 degrees |
| 9 | Final heading adjustment. |

to adjust travel distances. Sensor data are sent back at a rate of 20 readings per second to the computer. This allows students to review real-time data that give the robot's position and orientation, which can be graphed and analyzed in a variety of ways. A graph displaying the real-time position and orientation of the drivetrain is provided. Data from multiple trials can be saved and graphed simultaneously giving a clear graphical indication of the drivetrain repeatability, as shown in Fig. 12C, where the position of the drivetrain completing the square path shown in Fig. 12B during eight trial runs are graphed simultaneously.

Beyond the visualization of trajectories, a jupyter notebook is provided which can be used to evaluate the repeatability using saved data in more detail. The VEX code that runs

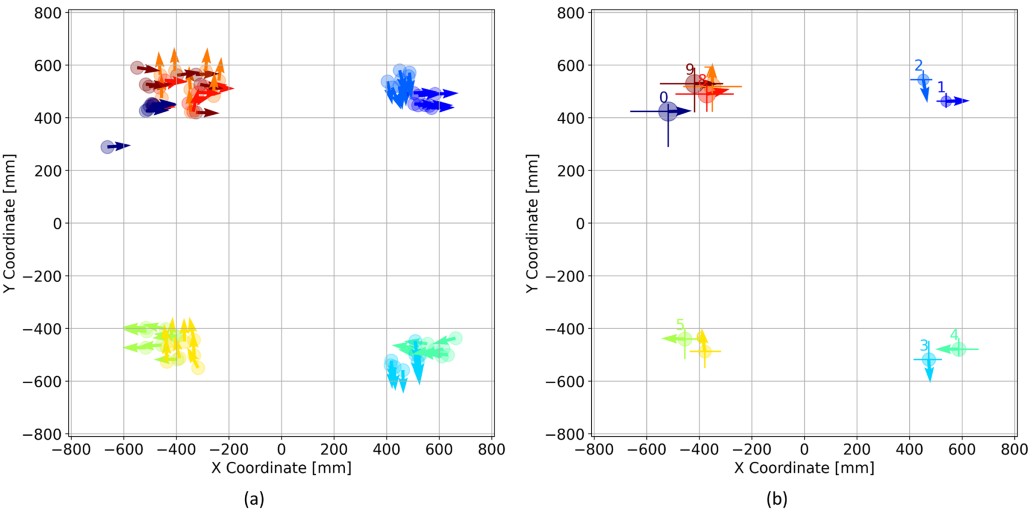

**Figure 13** (A) Drivetrain position and orientation at the endpoint of each trajectory segment shown for all 8 trials. (B) Drivetrain position and orientation averaged over 8 trials. The min and max values of the drivetrain x and y position are shown with the error bars.

**Table 8 Table of segment endpoints.** $x$, $y$, and $\theta$ are the robot field coordinates at the end of each movement segment, respectively.

| $S_{ID}$ | x Mean (stdev) [mm] | y mean (stdev) [mm] | $\theta$ mean (stdev) [degree] | x range $[min, max]$ [mm] | y Range $[min, max]$ [mm] | $\theta$ Range $[min, max]$ [degree] |
|---|---|---|---|---|---|---|
| 0 | −518 (55) | 423 (51) | 88 (0) | [−662, −486] | [289, 452] | [86, 89] |
| 1 | 539 (29) | 462 (21) | 88 (1) | [503, 584] | [437, 494] | [86, 91] |
| 2 | 452 (27) | 544 (21) | 170 (5) | [403, 487] | [515, 580] | [158, 179] |
| 3 | 473 (43) | −519 (32) | 176 (1) | [415, 523] | [−558, −448] | [174, 179] |
| 4 | 587 (49) | −479 (21) | 269 (10) | [509, 662] | [−506, −437] | [256, 281] |
| 5 | −455 (50) | −441 (38) | 271 (4) | [−516, −396] | [−517, −398] | [265, 281] |
| 6 | −379 (45) | −488 (38) | 353 (9) | [−437, −317] | [−550, −444] | [−28, 4] |
| 7 | −350 (80) | 518 (52) | 0 (14) | [−461, −238] | [422, 578] | [−4, 6] |
| 8 | −372 (78) | 489 (38) | 77 (25) | [−490, −268] | [423, 542] | [13, 97] |
| 9 | −418 (95) | 529 (50) | 91 (6) | [−549, −309] | [420, 590] | [82, 100] |

the drivetrain is built so that each segment of a trajectory is labeled independently. For example, segment 1 involves moving the drivetrain along a straight path for a given distance, while segment 2 involves turning the robot to the right by 90 degrees. The position and orientation of the drivetrain at the endpoint of each segment is plotted in Fig. 13A for all eight trials. Figure 13B displays the position and orientation of the drivetrain at the end of each segment averaged over all eight trials, as well as an error bar that indicates the maximum and minimum position. These data can be used to assess the reliability and repeatability of the drivetrain and compare the performance of design changes. The provided jupyter notebook can also be used to tabulate all of the trajectory averages as shown in Table 8.

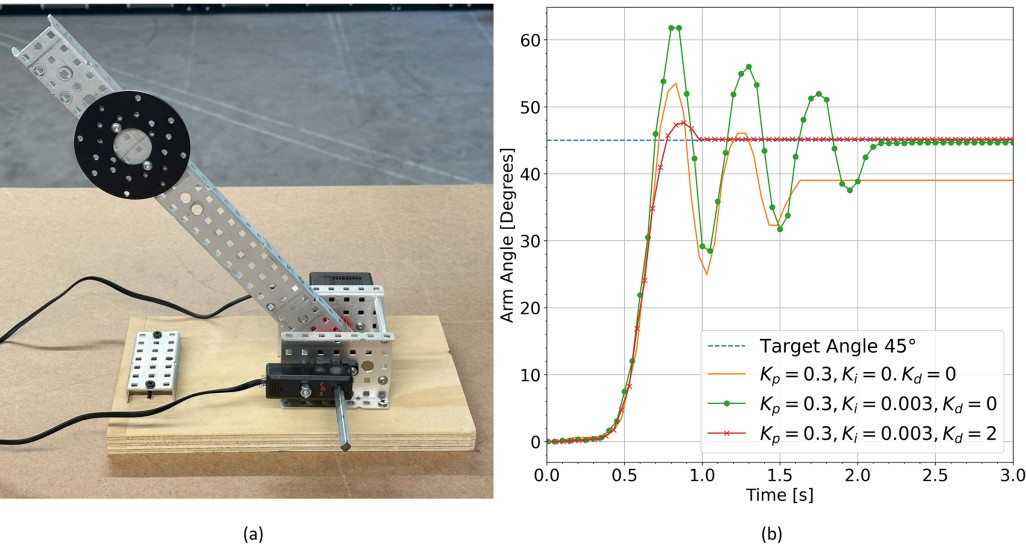

**Figure 14** **(A) Photograph of the weighted arm system. (B) System response given different PID parameters.**

## Tuning a PID controller

A proportional-integral-derivative controller, or PID controller, is a control feedback loop that corrects for deviations from the desired value. PID controllers are commonly used to precisely control parameters such as temperature, speed, or pressure. In robotics, PID controllers can be used to control the velocities of a drivetrain, for lifting objects, and for spinning flywheels. For example, the angular velocity of a flywheel might need to be precisely controlled to propel a projectile a particular distance. The controller would adjust the flywheel motor power as a function of the difference between the target flywheel angular velocity and the actual angular velocity. Here the angular velocity could be determined *via* a rotation sensor on the flywheel axle.

Another common example would be lifting an object to a desired height. The experimental setup shown in Fig. 14A illustrates the case where an arm has been fitted with a circular disc weight. The arm is driven directly through a motor at the base. A rotation sensor is attached to the axle passing through the arm and motor to monitor the arm configuration. The arm is to be lifted to a particular target angle. Motor power can be determined using a PID controller that responds to the difference, $e(t)$, between the target arm angle, $\theta_o$ and the actual arm angle measured by the rotation sensor, $\theta(t)$ where the difference is defined by:

$$e(t) = \theta(t) - \theta_o \tag{6}$$

A PID controller determines the output response as a function of a term that is proportional to the difference between the target and the actual value, a term that is proportional to the integral of this difference, and a term that is proportional to the derivative of this difference.

**Peer**J Computer Science

The equation for the instantaneous motor output, $u(t)$, is then defined as follows:

$$u(t) = K_p e(t) + K_i \int e(t)dt + K_d \frac{de(t)}{dt} \tag{7}$$

where $K_p$ is the coefficient of the proportional term, $K_i$ is the coefficient of the integral term and $K_d$ is the coefficient of the derivative term.

Each of the three coefficients $K_p$, $K_i$, and $K_d$ must be found for each application independently, as the optimal values depend on the characteristics of the system being driven. This process is known as tuning, and is often done by entering approximate coefficient values into the equation, observing the system response, and then further refining the coefficients.

Tuning is greatly simplified when real time data logging is possible. Students can observe the response of the system as they modify the PID parameters to determine if a particular set of coefficients is effective or not. Figure 14B displays the system response with various PID parameters and a target angle of 45 degrees.

Real time data logging shows much more accurately whether a particular set of coefficients result in an effective controller. Figure 14B illustrates that a controller with only a proportional term never reaches the target angle for this system. The proportional component of the controller increases the response when error values are high and decreases the response when error values are low according to tuned coefficients. Although the proportional component alone is an effective controller in many cases, it is not suitable for lifting applications where the output power may never be large enough to hold the weight at the target angle. The addition of an integral term allows the system to settle near the target value, and the addition of a derivative term eliminates the oscillations by slowing down the response when the measured value changes too quickly. This allows the system to settle at the target value much more quickly. All of this information is captured visually as the student modifies parameters of the PID model.

## CONCLUSIONS

In this article, we have created an educational platform that can be used to teach a multitude of advanced topics related to robotics. We demonstrated that the ESP-NOW protocol can be used to extend the functionality of the VEX V5 microprocessor to allow real-time object detection and real-time data logging. Further, the described system provides the ability to interface sensors and motors outside the VEX ecosystem, enhancing their capabilities for experimentation. Several example applications were presented, including object detection using YoloV8, reliability testing and trajectory analysis, and the tuning of PID controllers.The system can wirelessly log sensor data from the VEX V5 microprocessor in real time at a rate of 20 samples per second reliably to distances of 40 m from the transmitter. The transmission distance can probably be increased with a better placement of the transmitter relative to the ground and an additional distance between the transmitter and the surrounding metallic components of the drivetrain.

Object detection in real time was achieved using the YoloV8 algorithm. Round trip messages from computer to the VEX V5 microprocessor require 187 ms, and total prediction times for the YoloV8 algorithm range from 10.8 to 64.4 ms depending on the hardware used. Therefore, four to five sequential object identifications can be made per second using this system. The current speeds are sufficient to enable students to build real-time object manipulation applications. Future work will focus on optimizing the transmission protocol to reduce the round trip message time further.

The YoloV8 algorithm, trained on a dataset of 116 images of VEX game pieces, was also shown to be able to identify game pieces in a sample sorting application. This methodology can easily be extended to create robots that seek out particular objects, avoid objects, or manipulate objects depending on object identification. It is also straightforward to extend the methodology to larger groups of object types that are more difficult to distinguish from one another by increasing the number of training images used.

The real-time wireless data logging enabled by this system also allows the VEX V5 microprocessor to be used for more advanced experimentation in the classroom. Students can compute statistics related to repeatability as was done with trajectory analysis. It was also demonstrated that the difficult task of tuning a PID controller can be tackled visually by students using graphs generated using the developed Python GUI.

The low cost of the system, the ease of construction, and the software supplied in this study make the ESP-NOW enabled system an excellent choice for schools that have existing VEX equipment and a desire to work with ML or conduct more advanced experimentation.

### Funding
The authors received no funding for this work.

### Competing Interests
The authors declare that they have no competing interests.

### Author Contributions
- Emma I. Capaldi conceived and designed the experiments, performed the experiments, analyzed the data, performed the computation work, prepared figures and/or tables, authored or reviewed drafts of the article, and approved the final draft.

### Data Availability
The code running the system, results, and the raw data is available in the Supplemental Files.

### Supplemental Information
Supplemental information for this article can be found online at http://dx.doi.org/10.7717/peerj-cs.1826#supplemental-information.

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
