# Peer review of "A low-cost wireless extension for object detection and data logging for educational robotics using the ESP-NOW protocol"

_PeerJ Computer Science, doi:10.7717/peerj-cs.1826_

## Round 0.1 · original submission · Major Revisions

The paper must be improved taking into consideration all suggestions from reviewers.

·

Basic reporting

The manuscript is written in English. The manuscript is well written. It is clear and unambiguous. The manuscript includes sufficient introduction and background. The manuscript is well structured. Figures have good resolution and are appropriately described and labeled. The manuscript comes accompanied with electronic files. It is self-contained. Theorems are not included since it tries merely with the evaluation of some platforms through the use of vision (object detection) algorithms and sensors and their interconnectivity with the VEX V5 microprocessor from VEX Robotics system.

Experimental design

The manuscript meets with the aims and scope of the PeerJ Computer Science. The research question is well defined and meaningful. The investigation is conducted rigorously and to a high technical standard. Methods are described in sufficient detail and information to replicate.

Validity of the findings

All the results seem that can be replicated. These are accessible, data and electronic files have been attached to this submission and the robotics system is commercially available. Conclusions are well stated, linked to the original research question and limited to the supporting results.

Additional comments

There are a few grammar errors highlighted on the manuscript attached to this review; see pages 2/16 and 14/16.

Reviewer 2 ·

Basic reporting

The research focuses on the low-cost wireless extension for object detection and data logging for educational robotics using the ESP-NOW protocol. The authors have done excellent work. However, there are some advices for the manuscript.
1. Please clarify the main contributions of the research. Considering that “real time” is one of the core factors of the research, please provide exact data clearly to demonstrate the capability of “real time” of the system based on ESP-NOW protocol.
2. According to Table 2, the mean of the percent loss for 30 meters is much higher than that of 40 meters. Could you please provide more detailed explanations?
In addition, please provide more experimental details for the data shown in Table 2. In other words, how did you get the results?
3. The research involves wireless communication protocol ESP-NOW, object detection, PID controller, etc. The key idea of your proposal needs to be further clarified.
4. The conclusion needs to be refined with more precise words.

Experimental design

please make further experiments to demonstrate the capability of "real time" of the research.

Validity of the findings

no comment

Additional comments

no comment

Reviewer 3 ·

Basic reporting

- Line 28 : the reference is not complete in the text (J. et al., 2017).
- A rearrangement of figures location is suitable. Example figure 1 is cited in page 3 but displayed in the next page. Same thing for figure 2.
- Line 195 you cite figure 9a which is in page 11. Why it is not figure 3 and you can cite it in page 11 as figure 3.
- In line 205 you cite figure 5 before citing figure 4. If needed change the order.
- Line 329 e(t) was well defined before and no need to redefine it.
- Some references such as (Abdelkader et al., 2017) are not complete in the reference list.

Experimental design

- Table 2 : I think the first row is useful and can be deleted.
- In the section Tuning a PID Controller you discuss the advantage of the use of integral and the derivative term but you don’t mention the use of the proportional part which have an effect on the rising time.

Validity of the findings

- Table 2: what is the explanation for the result obtained for 40m.
- The results obtained and shown in figure 7 and 8 are predicted since the same objects are used for learning and test and the use of tiny dataset.

Additional comments

No other comments

---

## Round 0.2 · accepted · Accept

The paper can be accepted. It was improved well enough!

Reviewer 2 ·

Basic reporting

The current version is well organised and written. The manuscript clearly presents the work with sufficient literature references, professional article structure.

Experimental design

The experiments have been thoughtfully designed, with particular attention given to the inclusion of real-time performance data, which addresses my previous concerns. Consequently, I am satisfied with the provided responses and the overall experimental approach.

Validity of the findings

The necessary data has been fully presented in the manuscript.

Additional comments

The authors have made comprehensive revisions to the manuscript in accordance with the previous comments, and I am pleased to note that all of my concerns have been effectively addressed. Therefore, I am in agreement to accept the manuscript in its current form.